# Microparticles in the Development and Improvement of Pharmaceutical Formulations: An Analysis of In Vitro and In Vivo Studies

**DOI:** 10.3390/ijms24065441

**Published:** 2023-03-13

**Authors:** Rita Y. P. da Silva, Danielle L. B. de Menezes, Verônica da S. Oliveira, Attilio Converti, Ádley A. N. de Lima

**Affiliations:** 1Department of Pharmacy, Federal University of Rio Grande do Norte, Natal 59012-570, RN, Brazil; 2Department of Civil, Chemical and Environment Engineering, Pole of Chemical Engineering, University of Genoa, I-16145 Genoa, Italy

**Keywords:** polymeric microparticles, biological activity, in vitro assay, in vivo assay, drug delivery systems

## Abstract

Microparticulate systems such as microparticles, microspheres, microcapsules or any particle in a micrometer scale (usually of 1–1000 µm) are widely used as drug delivery systems, because they offer higher therapeutic and diagnostic performance compared to conventional drug delivery forms. These systems can be manufactured with many raw materials, especially polymers, most of which have been effective in improving the physicochemical properties and biological activities of active compounds. This review will focus on the in vivo and in vitro application in the last decade (2012 to 2022) of different active pharmaceutical ingredients microencapsulated in polymeric or lipid matrices, the main formulation factors (excipients and techniques) and mostly their biological activities, with the aim of introducing and discussing the potential applicability of microparticulate systems in the pharmaceutical field.

## 1. Introduction

The application of technologies in micro- and nanoscales to develop new therapeutic alternatives has grown considerably in the pharmaceutical industry. Polymeric systems for controlled drug release can be developed within these particle size scales [1,2]. Structurally, these systems differ in their size scale, i.e., microscale for particle diameter between 1 and 1000 µm [3] and nanoscale between 1 and 100 nm [4]. In addition to the size difference, there is a range of other properties where micro- and nanoparticles differ, such as crystallization, solubility, melting point, vitreous transition temperature, dissolution, etc., which allow them to be used in different applications [5].

Among the advantages of microparticles over nanoparticles we can cite the fact that, when the size is greater than 100 nm, they not cross the interstitium when transported by the lymph, thus having stronger local effect [3], and that they have better retention profile in the skin [6]. Another advantage refers to the method of pulmonary administration route, since particles with a diameter of less than 10 µm can arrive in the pulmonary alveolar region, where tissue permeability is high and gas exchange occurs. In addition, particles less than 20 µm, when reaching the bloodstream, can be phagocyted by macrophages [7]. Due to their anatomical differences, animal models may not reflect reliably what occurs when humans are exposed to these particles [8].

Structurally, microparticles are divided into two large groups, that is, microspheres, in which the active compound and the raw material are dispersed or dissolved homogeneously, and microcapsules, in which there is a membrane enclosure delimiting and encompassing the nucleus—solid, liquid or gaseous—where the active principle is deposited [3,9]. Other variants of microparticles can be manufactured with different multilayers, nuclei or irregular shapes (Figure 1) [10].

The particularities of these systems in relation to size distribution, morphology, composition and functionality depend directly on factors such as (1) physicochemical characteristics of the encapsulated active principle, (2) pharmaceutical form and administration method, (3) structural components (synthetic, semi-synthetic and natural polymers, lipids, among others) and (4) production method (physical, physicochemical or chemical). Microparticulated systems can be applied in different pharmaceutical forms, namely in solid (tablets, capsules and micropellets), liquid (suspensions and parenteral formulations) or semi-solid (gels, pastes, creams) state; therefore, they can be administered orally, pulmonarily, intravenously, intramuscularly and/or subcutaneously, depending on their size and release properties [11]. Attributes such as controlled, sustained and specific release to cells and/or tissues can be conferred to microparticles. It is also possible to administer poorly soluble drugs [12], to overcome epithelial and endothelial barriers [13], co-administer two or more bioactive agents for combined therapy [14], select adjuvants for improving the immune response of vaccines [15], and increase effectiveness, safety and bioavailability of therapeutic agents [16].

This review, by discussing the results of in vitro and in vivo tests, explores features improved in bioactive compounds by microencapsulation. This review, through the discussion of articles on in vitro and in vivo tests, addresses aspects that were improved in bioactive compounds through microencapsulation. The formation of microparticles through different methods and addition of structural components to compose the matrix, can provide changes in the physicochemical properties of bioactive compounds, which can be due to interactions between the constituents. In addition, the reduction in particle size can also favor increases in solubility, bioavailability and cell permeation, thus facilitating passage through biological membranes and enabling other biological processes such as excretion. The review also brings the current scenario of the market, citing some already marketed drugs that use microparticulate systems and their benefits compared to conventional formulations.

## 2. Methodology

Bibliographic search was conducted in specialized databases (ScienceDirect, Scopus and PubMed) using different combinations of the terms “polymeric microparticles”, “microparticles”, “microspheres” and “microcapsules” and taking into account the biological activity of interest, namely anticancer, anti-inflammatory, antibacterial, antiparasitic, antioxidant or wound healing activity. The inclusion criterion used was in vitro or in vivo studies where the action of active compounds loaded in polymeric microparticles was investigated.

## 3. Polymeric Microparticles in Biological Tests

The in vitro and in vivo analyses are preclinical tests used to evaluate the effectiveness and safety of pharmacologically active compounds before clinical trials in humans. For microencapsulated systems, in vitro tests on cell cultures and in vivo tests in mice are generally used to evaluate the influence of encapsulation on the biological properties of active ingredients (Figure 2).

### 3.1. Anticancer Activity

The therapeutic use of systemic drugs in cancer treatment has serious side effects and results in low cure rates. The use of microencapsulation techniques for the preparation of targeted drug release systems has been well studied as a strategy to increase efficacy and reduce agents’ toxicity [17,18,19].

#### 3.1.1. In Vitro Anticancer Activity

Ferulic acid (FA) is known for having anticancer and antioxidant activity and no toxicity to normal cells even at high doses [18,19]. Johnson et al. [20] included ferulic acid in fructooligosaccharide (FA FOS I) by self-assembly in microparticles for targeted release in the colon, whose effect was analyzed in human colon cancer cell lines HT-29 and LoVo and compared to that of normal colon fibroblast CCD18-Co by the WST-8 test. Not only the in vitro analysis against cancer cells showed greater selectivity of FA FOS I than oxaliplatin as a reference drug and free FA, but its lower half-maximal inhibitory concentration (IC_50_) values compared to free FA in the two tested cell lines also indicated the potentiality of the biological effect.

In another study, Çetin and Gümüşderelioğlu et al. [21] developed an implantable 3D skeleton-based system to combine osteoinductive and anticarcinogenic properties of melatonin. Two melatonin transport and release systems were prepared and incorporated into porous chitosan/hydroxyapatite (HAp) structures: (1) poly(lactic-co-glycolic acid) (PLGA) microparticles loaded with melatonin to stimulate bone regeneration with sustained release and to inhibit osteosarcoma cells and (2) a melatonin/cyclodextrin inclusion complex to quickly release large amount of melatonin. To investigate the osteoinductive properties of melatonin-carrying microparticle support systems, tests were carried out on the MC3T3-E1 mouse osteoblastic cell line for 21 days, while the anticarcinogenic properties of these systems and chitosan-based skeletons with inclusion complexes were tested on human osteosarcoma cells MG-63. The results indicated that the melatonin-release chitosan/HAp support, carrying high and low quantity-release systems (melatonin/HPβCD inclusion complex and PLGA microparticles, respectively), is a suitable system for human osteosarcoma treatment.

The use of gene therapy to induce wild-type p53 gene expression associated with small molecules was proposed by Shi et al. [22], who developed porous PLGA microparticles carrying doxorubicin and a plasmid containing the P53 gene by the solvent evaporation method in W/O/W emulsion. In the in vitro analysis performed in human lung adenocarcinoma cells A549 for 7 days, microparticles containing doxorubicin and PEI25K/p53 showed greater tumor growth inhibition and apoptosis induction than those loaded with any of the isolated agents.

#### 3.1.2. In Vivo Anticancer Activity

Oridonin is an active diterpenoid isolated from a plant used in traditional Chinese medicine that has anticancer effects and few adverse reactions. Zhu et al. [23] presented a porous microparticle (LPMP) carrying oridonin for the in situ treatment of primary non-small cell lung cancer (NSCLC) through pulmonary release. The PLGA LPMP system loaded with oridonin showed high anti-NSCLC effect on tested rats due to direct action on cancer cells after pulmonary delivery.

Delivery of vaccine antigens with appropriate systems can trigger intense immune responses against cancer, leading to reduced tumor growth and improved survival. Joshi et al. [24] developed microparticles using 1,8-bis(*p*-carboxyphenoxy)-3,6-dioxaoctane (CPTEG) and 1,6-bis(*p*-carboxyphenoxy)hexane (CPH), compounds with proven antigen-carrying ability, immunogenicity and antitumor activity. Mice were vaccinated with 50:50 CPTEG:CPH microparticles encapsulating a model tumor antigen, ovalbumin (OVA), in combination with the Toll-like-9 receptor agonist, oligonucleotide CpG 1826 (CpG ODN), which triggered higher TCD8+ cell responses after 14 and 20 days, compared to other treatment groups. In addition, microparticles containing OVA and CpG ODN generated more intense anti-OVA IgG1 antibodies response, but mice showed no significant protection against the tumor.

The association of two or more drugs to improve the treatment of cancer patients can be a challenge, as interactions between drugs may occur. To associate paclitaxel (PTX) and imatinib (IMN), Liu et al. [25] prepared polymeric microcapsules with external enclosure of PTX-loading hyaluronate and IMN-loading PLGA nucleus, to be vaginally administered for cervical cancer treatment. The in vitro release study showed a two-step release pattern: in the former step approximately 80% of PTX contained in the outer layer was quickly released, while in the latter, of the sustained type, approximately 90% of IMN was released within 7 days. Cytotoxicity tests in cell culture showed synergistic effects and much lower IC_50_ values for microencapsulated drugs than for the free ones. Vaginal retention time was prolonged, probably due to adhesion provided by hyaluronate. In in vivo test microparticles showed a much higher tumor inhibition rate (90%) than the mixture of free drugs at the same dose. Vaginal mucosa morphology of the microparticle-treated group was, after treatment, similar to that of the healthy group, showing the effectiveness of microparticles to improve cervical cancer treatment by reducing dose and adverse effects.

On the other hand, to improve ovary cancer chemotherapy, Dwivedi et al. [26] developed PTX-loading hybrid microparticles (PTX-HyB-MPs), whose anticancer effect was evaluated in vitro for 72 h in SKOV-3 ovary cancer cells and in vivo using a cancer xenotransplant model in female mice (20 mg/kg PTX equivalent dose), and compared to that of conventional drug delivery methods. The results showed that PTX-HyB-MPs can be potentially used for locoregional treatment of ovary cancer and other malignant tissue diseases.

Karan et al. [27] formulated 5-fluorouracil (5-FU)-loading microspheres by the emulsion solvent evaporation method to get a formulation for targeted controlled oral anticancer drug administration. Their in vitro activity was tested in breast cancer cells (MDA-MB 231) at concentrations of 5, 10 and 20 µg/mL and compared to that of 5-FU alone. Cytotoxicity of drug and 5-FU-loading microspheres proved to be dose-dependent, and the 5-FU-treated groups showed more intense apoptosis characteristics. Regarding the in vivo test, a liquid tumor model was performed by intraperitoneal inoculation of EAC cells in the peritoneal cavity of mice, and drugs were administered for 14 days. Microencapsulated 5-FU formulations were better tolerated than the compound alone, probably because they allowed prolonged release and less toxicity.

Vincristine (VCR) is a compound used in the treatment of lung cancer. To improve its effectiveness, Xu et al. [28] prepared VCR-liposomes that were then dried by spraying with excipients to produce dry powders for pulmonary distribution. In vitro cytotoxicity was tested by the MTT assay in MCF-7 and A549 cells, and in vivo experiments were performed in male Sprague–Dawley rats. Compared to the VCR solution, both VCR liposomes and their dry powders showed better ability to kill cells, indicating better antiproliferative activities. In addition, VCR pulmonary administration reduced liver metabolism, making the drug bioavailable longer. The maximum concentration (*C*_max_) of fry powders was 6.3 higher than those of the VCR solution and half-life (*t*_1/2_) was 81.1% higher, while the clearance was reduced by 83.2%. The results suggest that the proposed formulation is adequate to treat lung disease thanks to better pharmacokinetic properties.

Kim et al. [29] developed a microparticle system for inhalable sustained-release based on co-distribution of tumor necrosis factor-related apoptosis-inducing ligand (TRAIL) and doxorubicin (DOX) for the alternative treatment of metastatic lung cancer. They then evaluated the in vitro synergistic cytotoxicity to H226 cells, pulmonary deposition characteristics and antitumor efficacy of this distribution system in a mouse model of metastatic pulmonary carcinoma using H226 cells. Pulmonary administration of TRAIL/DOX PLGA microparticles resulted in their deposition in mouse lungs and their stay in situ for up to a week. Tumors in mice implanted with H226 cells and treated with the microparticles were markedly smaller in size and number than in mice treated with TRAIL or DOX PLGA alone.

In 2013, Madan et al. [30] evaluated in vitro and in vivo the physicochemical properties of sterically stabilized gelatin microassemblies of noscapine (SSGMs) to target human non-small cell lung cancer A549 cells. In vitro cytotoxicity tests against A549 cells showed for SSGMs a much lower IC_50_ value (15.5 µM) than those of gelatin (30.1 µM) and noscapine (47.2 µM) alone. Caspase-3 activity suggested that SSGMs could have enhanced the therapeutic effect of nascapine through prolonged release. The in vivo tests were performed in Swiss albinos male mice by administering gelatin and noscapine alone as well as nascapine-loading SSGMs at a dosage of 50 mg/kg body weight through the tail vein. Pharmacokinetic and biodistribution analysis showed that SSGMs increased the plasma half-life of noscapine by 9.57 times with a 48% drug accumulation in the lungs.

### 3.2. Antiparasitic Activity

Most antiparasitic drugs have low bioavailability due to their insolubility and short half-life, requiring frequent dosage due to long-life cycles of parasites [31], but some studies have shown the use of polymeric microparticles as a strategy to circumvent these limitations.

Leishmaniases, diseases caused by different protozoan species belonging to the *Leishmania* genus, can be divided into two groups based on clinical manifestations: cutaneous leishmaniasis (CL) and visceral leishmaniasis (VL). In 2020, Allotey-Babington et al. [32] prepared microparticles loaded with quinine sulfate in bovine serum albumin by spray drying and evaluated their effect on peritoneal macrophages of *Leishmania donovani*-infected rats. Infected macrophages received various treatments with culture media containing twice the IC_50_ of amphotericin B and 6.25 µg/mL microparticles and quinine sulfate. The reduction of parasitic load in groups treated with microparticles of quinine sulfate and amphotericin B was much larger than in the quinine sulfate-treated group. In order to determine the pharmacokinetic parameters, groups of male Sprag-Dawley rats received, separately, 30 mg/kg microparticle formulation and 3 mg/kg amphotericin B, intraperitoneally, and later blood and tissue samples were collected in specific points. *C*_max_, *t*_1/2_ and area under the curve (AUC)—area under the plot of plasma concentration of a drug versus time after dosage—were all significantly better for quinine sulfate microparticles compared to free drug. The formulation of microparticles reduced the concentration of *L. donovani* in the blood and tissues (liver and spleen) of infected rats more effectively than amphotericin B used in clinical practice.

Aiming at a treatment for VL, Collier et al. [33] proposed a formulation with acetalated dextran microparticles carrying 2-amino-N-[4-[5-(2 phenanthrenyl)-3-(trifluoromethyl)-1H-pyrazol-1-yl]phenyl]-acetamide (AR-12/MPs). For in vitro leishmanicide activity, bone marrow macrophages were infected with *L. donovani* promastigotes in the 1:7 ratio and treated with samples for 24 and 72 h. AR-12/MPs at 0.5, 1.0 and 2.5 mM concentrations suppressed intracellular amastigotes after 24 h by 70, 83 and 90%, respectively, while under the same conditions AR-12 alone yielded only 68, 52 and 80%, respectively. After 72 h, microparticles still had greater leishmanicide activity, confirming that the system enhanced the drug effect.

In 2019, to treat CL Sousa-Batista et al. [34] designed PLGA microparticles loaded with amphotericin B deoxycholate (d-AmB) as a topical formulation for macrophage intracellular targeting and sustained extracellular delivery. For in vitro anti-amastigote activity, mouse peritoneal macrophages were incubated with *Leishmania amazonensis* promastigotes. Infected macrophages were treated with d-AmB and d-AmB/PLGA at equivalent AmB concentrations (0.01–100 μm) or PLGA for 48 h to 37 °C. The anti-amastigote activity of treatment with d-AmB/PLGA was similar to that of treatment with d-AmB (IC_50_ = 0.05 μg/mL versus 0.08 μg/mL), but the selectivity index of microparticles (SI = 50) was twice that of the drug alone. For in vivo analysis, BALB/c mice were infected in the ear with 2 × 10^6^
*L. amazonensis* promastigotes and, after 10 and 30 days, were treated intralesionally with d-AmB, d-AmB/PLGA or AmBisome Liposomal (5 μg AmB/10 μL PBS, 0.2 mg/kg). Treatment of initial injuries with a single injection of d-AmB/PLGA made 10 days after the infection was more effective in slowing (by at least 80 days) parasite growth than with eight free d-AmB injections in 120 days. Indeed, the lesions were 37% smaller, and the parasite load in the ear tissue and lymph nodes was 85 and 78%, respectively. Pharmacokinetic study showed that, after d-AmB/PLGA injection, AmB leaked more slowly in uninfected than infected ears, still remaining in the ear tissue for up to 30 days. Interestingly, AmB was undetectable in circulating plasma for at least two weeks of d-AmB/PLGA injection.

Still aiming at a topical CL treatment, Sousa-Batista et al. [35] produced PLGA microparticles loaded with CH8 (an antileishmanial nitrochalcone) that circumvented skin permeation and promoted drug absorption by infected macrophages and sustained local release. In vitro tests of microparticle capture by macrophage and anti-amastigote cytotoxicity showed that CH8 microencapsulation improved its safety and maintained its high selectivity index due to the direct action of intracellularly released drug rather than macrophage activation. For in vivo analysis, mice were infected in the ear pinna with *L. amazonensis* promastigotes. Triple intralesional injection of CH8/PLGA reduced the parasitic load by 97%, single injection by 91%, while standard intralesional CL treatment by only 36%.

In another study, Sousa-Batista et al. [36] prepared CH8 analogues to develop PLGA microparticles and evaluated their antileishmanial activity in vitro and in vivo. Results similar to those mentioned above were found for PLGA/polyvinylpyrolidone (PVP) microspheres loaded with the NAT22 analog. The in vitro activity test showed for NAT22-PLGA a capability of killing the parasite intracellularly (IC_50_~0.2 µM) similar to that of free NAT22 alone, while in vivo studies, conducted in *L. amazonensis*-infected mice, demonstrated significantly higher efficacy of NAT22-PLGA in reducing the parasitic load with a single intralesional injection.

Due to their low water solubility, imidazozoquinolines are mainly restricted to topical formulations and CL treatment. To overcome this obstacle, Duong et al. [37] encapsulated the imidazoquinoline resiquimod in acetalated dextran by the electrospray method and proposed a parenteral formulation to treat VL. In vitro cytotoxicity was tested in murine RAW 264.7 macrophages treated with microparticles alone, loaded microparticles and free drug, and the productions of nitric oxide and inflammatory cytokines were determined. The treatment with loaded microparticles induced a significantly higher immune response in macrophages compared to the free drug. In vivo, BALB/c mice were infected with *L. donovani* amastigotes by injection into the tail vein and treated intravenously through a single 100 μL injection of PBS, empty microparticles (1 mg MPs) or drug-loaded microparticles (1 mg MPs, 55 μg resiquimod) 14 days after infection. The parasite load was significantly reduced in the bone marrow (~40 amastigotes/200 macrophages) compared to blank particles and controls, which indicates improvement of drug delivery by microencapsulation.

Hoseini et al. [38] conducted a comparative study in vivo between chitin and chitosan microparticles in order to evaluate possible immunomodulatory activity of these compounds in BALB/c mice infected with *Leishmania major*. Animals were treated with these MPs (100 µg/100 µL) separately by subcutaneous injection every two days for two weeks, using PBS for the control group. Viable parasites were determined by the limiting dilution test 12 weeks after infection. After recovery of lymph node cells, cytokine production and lesion size were evaluated. Skin lesion was reduced in groups treated with chitosan-MPs (0.6 ± 0.12 mm) and chitin-MPs (1.2 ± 0.8 mm), when compared to control (6.2 ± 1.7 mm). The parasite load in the lymphatic ganglia was 1.31 × 10^6^, 7.49 × 10^6^ and 8.24 × 10^7^ parasites/lymphatic nodule for chitin-MPs, chitosan and control, respectively. Chitin-MPs also had greater IL-10 expression compared to chitosan.

Ospina villa et al. [39] developed PLGA microparticles loaded with two *Leishmania panamensis* proteins, LpanUA.22.1260 e LpanUA.27.1860, in order to evaluate the in vivo protective effectiveness of these systems in BALB/c mice affected by leishmaniasis. For the in vivo test, three doses of 100 µL of PLGp-rLpanUA.27.1860, a protein expressed in the promastigote and amastigote phases, were injected into mice every two weeks. Nine weeks after the first injection, 1 × 10^5^
*L. panamensis* promastigotes were injected into the ears of animals, while the controls received injections of PBs and empty microparticles. The size and severity of lesions were determined over 8 weeks after infection, and the results showed that the mice treated with PLGp-rLpanUA.27.1860 had smaller lesions compared to control groups, with no adverse effects on body mass and general health of animals.

The development of noninvasive vaccines aims at producing prolonged immunity. In this context, Gomes et al. [40] prepared chitosan microparticles crosslinked with glyceraldehyde as an intranasal vaccine (with LACK-DNA plasmid) against *Leishmania infantum* and evaluated their effectiveness in vivo after 7 days, 3 months and 6 months. Rats that received vaccine showed a significant reduction in the parasitic load in the liver and spleen compared to the control group, even after 6 months. The use of chitosan as an adjuvant helped in the induction of humoral and cellular immunity.

Malaria is still the parasitic disease that kills more people in the world. Caused by *Plasmodium* sp., it is more common in tropical and subtropical regions and can evolve quickly and severely. The causative parasite has developed mechanisms of resistance to treatments employed over the years, and the currently available options have many adverse effects and need various daily administrations to achieve the therapeutic effect, which further worsens adverse effects [41]. Aiming to improve treatment proposals, da Silva de Barros et al. [42] encapsulated primaquine in polylactic acid (PLA) polymeric microparticles to fight the hepatic primary phase of the parasite. In vitro release studies showed a prolonged release of the drug, which allowed to reduce the number of administrations and adverse effects, while in vivo studies demonstrated long drug permanence in hepatocytes (74.55%).

### 3.3. Antibacterial Activity

The discovery of antibiotics has revolutionized the treatment of infectious diseases. Some of these drugs, however, posed problems related to poor solubility and bioavailability. Moreover, frequent and unnecessary exposure of bacteria to these compounds caused the development of bacterial resistance mechanisms, which have limited their use. The use of release systems to transport antibiotics, such as microparticulated systems, helps to circumvent resistance mechanisms and modulate their physicochemical characteristics. The application of these systems has been explored and proves to be a promising alternative in the solution of these problems [43,44,45].

#### 3.3.1. In Vitro Antibacterial Activity

Carbapenems are antimicrobials used against a wide range of Gram-positive, Gram-negative, aerobic and anaerobic bacteria. Doripenem, a drug belonging to this class, is widely used in the treatment of pneumonia and, similar to others of its class, has low oral bioavailability and is administered intravenously. Yildiz-Peköz et al. [46] proposed doripenem loading in chitosan microparticles, whose properties were optimized using lactose, trehalose and leucine in different concentrations. The in vitro test results showed that microparticles allowed controlled release considered suitable for drugs administered by the pulmonary route. The release was delayed with increasing concentrations of lactose, trehalose and leucine. The formulation also exhibited low cell toxicity, especially at lactose concentrations between 20–30%, and good antimicrobial activity.

Still, in order to develop formulations to improve the treatment of lung diseases, but this time with the objective of treating tuberculosis, Cunha et al. [47] loaded antimicrobials such as isoniazid and rifabutin in chitosan microparticles to test their in vitro activity. When the antibacterial activity was tested against *Mycobacterium bovis* BCG, it was found no reduction of the antibacterial potential of any of the drugs and even a decrease of the minimum inhibitory concentration (MIC) of isoniazid when used in association with rifabutin. The formulation showed an average particle size suitable for pulmonary deposition (4 µm), good fluidity and aerosolization properties and complete release within 2 h, with some variations as the pH was changed. These results show a great potential of the system to be applied as inhalation treatment for tuberculosis.

The ability of some bacteria to live in the intracellular environment of phagocytes protects them not only against the attack of the immune system, but also against the action of antibacterial agents. Most antibiotics have low ability to permeate cell membranes; therefore, their incorporation into delivery systems, such as microparticles, has been studied to improve direction to intracellular targets. In this context, Maghrebi et al. [48] proposed a hybrid PLGA and lipid system for rifampicin (Rif-PLH) sustained release and studied its intracellular delivery capacity. Small-colony variants of *Staphylococcus aureus* were used for in vitro tests. The results obtained with this system were compared with those of free drug and drug associated only with PLGA (Rif-PLGA). After one hour of incubation, <15% of free rifampicin was released into macrophages, compared to the 51 ± 2% dose of Rif-PLGA and 78 ± 4% of Rif-PLH, as shown in Figure 3. It was observed that the Rif-PLH system allowed for intracellular concentrations twice as high as Rif-PLGA, i.e., better permanence capacity within the host cell. In addition, treatment using Rif-PLH at MIC (0.50 µg/mL) reduced *S. aureus* viability four times more than the free drug at the same concentration, and the reduction of colony-forming units by Rif-PLH was twice higher than that by Rif-PLGA using the same dose.

Ali Said et al. [49] also made a study to target resistant bacteria that live in the intracellular environment. In their study, porous calcium carbonate (CaCO_3_) microparticles were studied in two aspects. The former dealt with the ability of CaCO_3_ to load antibiotics, namely penicillin, ampicillin and ciprofloxacin, while the latter with the use of CaCO_3_ as the basis for developing a carrier material with inherent antibacterial properties, using chitosan (which already has proven antibacterial action) and dextran to coat microparticles. Inhibition of *S. aureus* and *Escherichia coli* growth occurred 2 h after administrating microparticles loaded with antibiotics. This was surprising especially in the case of ciprofloxacin that is practically insoluble in aqueous medium and against which there are proven resistance mechanisms, although its effectiveness occurred only at the MIC. Chitosan-loaded microparticles had their CaCO_3_ core removed, which led to the formation of microcapsules with enhanced inhibitory effect against the two target bacteria in in vitro assays. This was later confirmed by the fluorescence test using acridine orange, which emitted red or green fluorescence when binding to the single-stranded denatured DNA of bacteria exposed to microcapsules or to double-stranded integral DNA of the non-exposed ones, respectively.

#### 3.3.2. In Vivo Antibacterial Activity

To obtain effervescent formulations for the pulmonary route, Wang et al. [50] prepared azithromycin (AZM)-loaded fumaryl diketopiperazine microparticles by spray drying. To evaluate the in vivo antibacterial efficiency of normal (MPs) and effervescent (E-MPs) microparticles, BALB/c mice were infected with a suspension containing *Streptococcus pneumoniae* and after 24 h received treatment. Mice blood samples were collected to follow IL-6 and IL-10 interleukin concentrations. In the group treated with E-MPs (75 mg/kg/day), IL-6 decreased quickly after 4 (60 pg/mL) to 8 (50 pg/mL) days and IL-10 increased after 2 (130 pg/mL) to 10 (170 pg/mL) days when compared to other groups treated with the same dosage, suggesting greater therapeutic efficacy of E-MPs. After animal sacrifice, pulmonary tissue was analyzed to determine the number of colony-forming units (CFU). The histopathological analysis revealed inflammatory lesions in all infected groups, which, however, progressively regressed since the second day in groups treated with MPs, E-MPs and AZM in PBS until the disappearance of infection symptoms after 6 days, suggesting that E-MPs had a therapeutic effect similar to other treatments even if administered less frequently. From the second day, all treatments had significant antibacterial effects, especially that with E-MP. The relative body weight in groups treated with MPs and AZM increased for ten days after administration compared to the first day, while the relative lung weight increased only in the pneumonia model from 0.8 to 1%, suggesting control of pulmonary bacterial infection. The results showed that E-MPs had therapeutic effect similar to the drug already used, but with less frequent administration.

An alternative to treat gastrointestinal infection caused by *Helicobacter pylori* was studied by Hao et al. [51] in order to prepare a sustained release system. In the study, metronidazole (MDZ), one of the drugs used in the treatment of this bacterium, was deposited by the electrospray method onto Eudragit RS porous microparticles, whose activity was tested in vitro and in vivo in New Zealand rabbits. Gamma scintigraphy evidenced that even 8 h after administration there was still drug in the stomach. In the in vitro bacterial inhibition study, free MDZ proved to be more effective than microparticles after 6 h, but the opposite occurred after 12 h, showing that the system allowed a controlled release and was effective against the bacterium for a longer time. In addition, the damage caused to the gastric mucosa was lower as well.

Metritis is a uterine infection associated with several bacteria that affects 20–40% of cows after parturition, causing damage to their well-being, health and reproduction. The multiplicity of pathogens makes treatment a challenge, and its failure rate is around 30%. Jeon et al. [52] prepared chitosan microparticles (CM) by ionic reticulation and tested them as an alternative for the treatment of metritis. In the in vitro analysis, it was observed that CM concentrations ≥0.2% were sufficient to inhibit growth, in the uterine fluid of cows with metritis, of the intrauterine *E. coli* pathogen present in all tested cows. In the in vivo analysis, it was observed that the group treated with CM no longer had, 5 days after treatment, pathogen living cells, contrary to the group treated with the antibiotic ceftiofur, in which there was only a reduction. It was also observed that, continually exposing the bacterium to CM, it did not develop resistance. After CM administration, the concentrations of other pathogenic bacteria identified in the cow uterus by the sequencing technique were reduced in greater proportion compared to treatment with ceftiofur.

Capreomycin sulfate (CS) is the second-line drug in the treatment of tuberculosis, which proved to be quite effective in fighting pathogens; however, it has several adverse effects and, being very soluble, has a limited encapsulation capacity in hydrophobic systems. Cambronero-Rojas et al. [53] developed two systems containing this drug. In the former, CS was allowed to react with sodium oleate to form capreomycin oleate (CO), which is less soluble than the drug. In the latter, the obtained CO was loaded in PLGA microparticles through spray drying and solvent emulsion evaporation. The in vitro release study showed that CS was dissolved quickly and completely after 8 h, while CO and CO-loading microparticles showed slower diffusion. Microparticles prepared by different techniques had similar diffusion rates, with total release after 26–28 days. The results corroborate with those obtained in the in vivo pharmacokinetic study, in which the free drug and CO showed *C*_max_ 1 and 24 h after administration, while encapsulated CO a supported release for at least 8 days. The proposed system led to a satisfactory intramuscular deposit and was effective in reducing the number of doses to be administered.

Of the mentioned studies, 50% used chitosan as a polymeric matrix, probably due to its intrinsic antibacterial action. Table 1 shows the main information on other studies dealing with polymeric microparticles that used chitosan as a matrix and explored its antibacterial action to obtain synergistic effects as well as the typical advantages of drung release systems.

### 3.4. Antioxidant Activity

Reactive oxygen species (ROS), when produced in excess, have cytotoxic effects and alter normal cell function by unbalancing the endogenous antioxidant system [63]. Antioxidant substances have several clinical applications, such as anti-aging, anticancer, antidiabetic, anti-inflammatory, cardiovascular disease prevention, hepatoprotective, nephroprotective and neuroprotective actions, among others [64]. However, their use may be limited by their chemical instability and other physicochemical properties, such as low solubility. Therefore, release systems, such as microparticles, are an alternative that has been increasingly studied over the years and has proven to be effective [65].

Pheophytin A, a very common chlorophyll in edible green plants with high antioxidant activity but low aqueous solubility, was loaded by Mohammed et al. [66] in polymeric ethylcellulose microparticles in an attempt to develop an absorption system able to preserve its antioxidant activity. The assay of inhibition of nitro-blue tetrazolium formation was used to quantify pheophytin A antioxidant activity, which proved similar to that of ascorbic acid used as a standard at a concentration equal to half of the MIC. In addition, microparticles were safe even at high doses and showed low cell toxicity and antioxidant activity similar to that of already-known powerful antioxidants, suggesting that they could be used for this purpose.

Mangiferin (MG) is a glucosylxanthone, naturally produced by several plant species, whose high antioxidant capacity is hindered by its low water solubility and bioavailability. Therefore, Liu et al. [67] proposed MG encapsulation in microparticles (MG-MPs) using the supercritical antisolvent process and N,N-dimethylformamide as a solvent. Solubility tests, performed in water, artificial gastric juice (AGJ) and artificial intestinal juice (AIJ), showed that in all media MG-MPs solubility was greater than that of MG alone. In the 2,2-diphenyl-1-picrylhydrazyl free radical (DPPH·) assay for antioxidant activity, MG-MPs had IC_50_ values smaller than vitamin C used as a standard and higher radical scavenging ability at the same concentration. In the assays of reducing power and 2,2′-azino-bis(3-ethylbenzothiazoline-6-sulfonic acid) radical cation (ABTS^+^·) scavenging, the antioxidant activity approached that of vitamin C as the dose was increased. In addition, the MG-MPs IC_50_ values were also lower than those of free MG in both assays, indicating that a lower concentration of the micronized compound was sufficient to achieve the same removal yield. The pharmacokinetic study conducted in rats showed that, after oral administration, microparticles allowed a higher plasma MG concentration in less time than MG alone, revealing better absorption rate. Therefore, micronization of MG improved its solubility and bioavailability.

Selenium is an element that participates in important biological functions such as reducing oxidative stress. Bai et al. [68] loaded selenium nanoparticles (SE-NPs) in chitosan/chitooligosaccharides microparticles, as shown in Figure 4. Although the antioxidant activity of nanoparticles has already been described, its low stability impairs oral absorption. The in vitro release test showed that Se-NPs were released from intact microparticles, without changing morphology and size, and remained stable at 4 °C for a month, without deposition. The antioxidant activity was evaluated by the DPPH· radical and ·O_2_^−^ radical anion scavenging activity assays, using vitamin C as a standard. Although the antioxidant capacity of microparticles was lower than that of vitamin C (1/10 by the former assay and 1/6 by the latter), it was still considered acceptable being much larger than that of sodium selenite. The in vivo toxicity test with Kunming mice allowed to determine an average lethal dose (LD_50_) of microparticles about 20 times higher than that of selenite, showing that the incorporation of Se-NPs into polymeric microparticles improved antioxidant activity, significantly reduced toxicity and increased the formulation stability.

Rosacea is a chronic inflammatory condition of the skin that is almost always related to the presence of ROS. Metronidazole, early mentioned for its antibacterial activity, is one of the drugs used to treat this condition, which also has antioxidant activity resulting from neutrophil modulation. However, when administered by the oral route, it has low bioavailability. Therefore, Sulstiawati et al. [6] proposed a topical gel formulation of this drug inserted in solid-lipid microparticles, which showed a sustained release profile (>98% in 24 h), while the same amount of free drug took just 2 h. The antioxidant activity test showed very close IC_50_ values for free and encapsulated drugs, indicating that the formulation did not affect the compound antioxidant activity and that its topic administration would be feasible.

### 3.5. Anti-Inflammatory Activity

Although anti-inflammatory drugs are among the most prescribed drugs in the world, most of them are insoluble in water, resulting in low bioavailability. Therefore, the insertion of such drugs in polymeric systems could be used to increase their solubility and to improve their physicochemical and biological properties [69].

#### 3.5.1. In Vitro Anti-Inflammatory Activity

In the context of inflammatory bowel diseases (IBDs), Leonard et al. [70] combined polymeric micro and nanoparticles to encapsulate the anti-inflammatory glucocorticoid drug budesonide to increase its adherence to inflamed regions of the colon, as the drug transit time is insufficient due to constant diarrhea. The formulation proved to be resistant to the acidic stomach environment, being able to release only a small amount of the drug at pH 1.8 and more than 90% under intestinal conditions in one hour. The system increased adhesion to the surface of inflammatory cells, a characteristic observed in an in vitro 3D IBD model that was evaluated by fluorescence microscopy. In addition, there was a reduction in the concentrations of inflammatory cytokines IL-8 and IL-1β down to normal values, showing that the system is not only stable at stomach pH, but is also able to release the drug in less acidic organs, such as the colon, and to promote satisfactory therapeutic effects.

Studies with ibuprofen-lading microparticles, produced by an anti-solvent crystallization technique [71] and pH change [72], aimed to improve the dissolution profile of this drug. Both studies used excipients and surfactants in formulations to homogenize particle size and assist in their formation. Afrose et al. [71] observed that ibuprofen incorporated into hydroxypropylmethylcellulose microparticles had a higher dissolution rate than the free drug. Ren et al. [72] also obtained higher dissolution rate of this drug by inserting it into microparticles by adding Tween 80 as a surfactant. After 10 min, 60% of ibuprofen contained in microparticles was dissolved in the presence of the surfactant and only 15% in its absence versus 20% of the drug alone.

Widely used to treat various diseases, methotrexate has a number of adverse effects on bone, kidney and liver cells, among others. In psoriasis, an autoimmune disease, this medicine is used together with acitretin to induce peripheral T-cell apoptosis, but both are cytotoxic [73]. In order to reduce their adverse effects, Meilanczyk et al. [74] proposed the development of polymeric microparticles of these drugs, whose activities were tested against various cell lines, including normal human dermal fibroblasts, keratinocytes and bronchial epithelial cells. In vitro tests almost showed no cytotoxicity, while histological studies performed 48 h after administration of the conjugated system showed normal keratinocytes without skin irritation, corroborating the results obtained in vitro. The system proved promising and may be administered in various routes and pharmaceutical forms.

Diacerein is an anthraquinone derivative with anti-inflammatory properties able to inhibit IL-1β cytokine actions, including in cartilaginous cells, but its low water solubility causes low bioavailability. For this reason, Gómez-Gaete et al. [75] proposed PLGA microparticles loaded with rhein, the active principle of diacerein, for intra-articular administration and osteoarthritis treatment. The formulation exhibited prolonged release profile, with 45% of the drug released in 24 h and 90% in 30 days, while the cell viability test showed low toxicity. The inhibition of proinflammatory cytokine IL-1β and ROS, which are related to the activation of some inflammatory pathways, was significant compared to those of control and empty microparticles. The results as a whole showed a prolonged release formulation that would allow to reduce the number of administrations, which is a very desired feature, considering that these diseases are chronic and their treatment is prolonged. Figure 5 shows some of the benefits of rhein due to the formation of microparticles.

Resveratrol is a polyphenolic secondary metabolite produced by some plant species that has caught the attention of researchers for its high anti-inflammatory potential in nasal diseases. However, it has low bioavailability and solubility, is photosensitive and is quickly metabolized [76]. In view of this, Martignoni et al. [77] developed two types of solid lipid microparticles, one of which coated with chitosan in order to improve the compound physicochemical properties and minimize its toxicity in the nasal mucosa. Both formulations showed good dissolution profile and no toxicity to the tested nasal cell line, suggesting that the system can be a viable alternative for the treatment of nasal diseases, taking into account that current treatments are often expensive and have several local and systemic adverse effects.

#### 3.5.2. In Vivo Anti-Inflammatory Activity

Thinking about the treatment of inflammatory airway diseases associated with oxidative stress, such as asthma, Jeong et al. [78] developed vanillyl alcohol-containing co-polyoxalate (PVAX) microparticles loaded with dexamethasone (DEX). The production of nitric oxide, a pro-inflammatory mediator, was fully suppressed by DEX-loaded microparticles at concentration of 100 µg/mL, while, to achieve the same result, a double PVAX concentration was needed. The results of histological examination showed anti-asthmatic activity of DEX-loaded microparticles as a result of the suppression of the production of inflammatory mediators and synergistic effects of the polymer and drug, both with anti-inflammatory activity.

IBDs are often chronic and mainly affect colon mucosa without permanent cure. Due to the antioxidant and anti-inflammatory potential of quercetin, Helmy et al. [79] prepared and tested different formulations of chitosan microparticles loaded with quercetin to effectively deliver it to the sick colon. The in vitro release profile of microparticles exhibited minimum loss of drug in the simulated gastric and intestinal media, demonstrating its rapid and selective release in the medium simulating colon under pathological conditions. In turn, in in vivo tests the therapeutic efficiency was evaluated by the acetic acid-induced colitis model in rabbit and, as a response, quercetin-loaded microparticles targeted to colon showed higher therapeutic outcome than the drug alone.

In the same context, to improve the unfavorable pharmacokinetic profile of mesalazine, an aminosalicylate widely used to treat mild to moderate IBDs and increase its anti-inflammatory effect by rectal administration, Palma et al. [80] microencapsulated this drug in chitosan particles exploring the bioadhesive characteristics of the polysaccharide. In vitro and in vivo experiments confirmed the significant mucoadhesive characteristic of the formulation, demonstrating its therapeutic efficacy at a drug concentration (13 mg/kg) equal to half of that of Asamax^®^ commercial formulation (26 mg/kg).

Tenoxicam (TNX) is a powerful drug used to relieve pain, inflammation, swelling and rigidity that accompany rheumatoid arthritis and osteoarthritis. In 2019, Khattab et al. [81] associated TNX with sesame oil and poly (DL-lactide) in a sustained microparticle release system. The objective of the study was to reduce the adverse effects of the drug and allow its application by the parenteral route to improve the patient adherence to treatment. The formulation enhanced anti-inflammatory effectiveness, with greater biocompatibility with body tissues than the control. In addition, in vivo tests confirmed the higher antiarthritic efficacy of TNX-loaded microparticles in the treatment of induced arthritis compared to the orally administered drug.

Even though intra-articular therapy has been preferred in the treatment of osteoarthritis, drugs administered by this rout are rapidly eliminated by synovial fluid through the associated lymphatic system. Trying to minimize this problem, Sangsuwan et al. [82] inserted flavopyridol into PLGA microparticles for intra-articular administration and evaluation of its effectiveness when applied through this administration route. Matrix metalloprotease (MMP) is the main protease involved in cartilage degradation, and therefore, its activity was used to measure the inflammation stage after lesion induction in rats. Microparticles not only reduced protease activity 2–3 days after injury but also showed controlled release profile, longer articular retention time and less drug accumulation in the liver and kidneys compared to free drug.

Nonsteroid anti-inflammatory drugs are usually used to treat inflammatory processes associated with snake bites. Based on the association of various side effects, for example gastric and duodenal ulcer, bleeding and renal failure, Ribeiro et al. [83] evaluated the anti-inflammatory and antinociceptive effects of *Morus nigra* ethanolic extract, as well as its activity, when incorporated into electrospun fibers and polymeric microparticles, on mouse paw lesions induced by *Bothrops jararacussu* venom. Treatment with extract-containing fibers and microparticles decreased the edema of paw 2 and 4 h after venom infiltration, respectively, and altered the extract pharmacokinetics in vivo.

### 3.6. Healing Activity

The wound healing process involves a sequence of molecular and cellular events that occur after a lesion to repair the injured tissue. The use of microparticulated systems as transporters of specific types of cells or drugs on the injured skin can facilitate rapid healing without scarring [84].

#### 3.6.1. In Vitro Healing Activity

*Moringa oleifera* is a tree whose leaves are rich in bioactive compounds. The leave alcoholic extract is capable of inhibiting the secretion of various pro-inflammatory markers, while inducing the production of anti-inflammatory cytokines such as IL-10, and has antibacterial activity. Aiming at a formulation capable of swelling and forming film around the wound, Pagano et al. [85] developed hydrogel microparticles loaded with *M. oleifera* leaf extract capable of ensuring a sustained release of active compounds. The antioxidant activity, tested because ROS contribute to the delay of the healing process, was considered satisfactory in protecting tissue. The in vitro wound healing test was done with microparticles in two different concentrations, which were both able to stimulate cell growth. Tests with pork skin samples showed that microparticles were capable of creating a gel layer and protecting the wound when contacting the exudate. In addition, they were also effective against common microorganisms in wound infections. These results as a whole suggest that polymeric microparticles could be a viable alternative for extract administration.

The low concentration of growth factors is one of the causes of inflammatory process prolongation, which causes insufficient angiogenesis and delays healing of wounds that become more susceptible to bacterial infections. Thinking about it, Zarubova et al. [86] microencapsulated the vascular endothelial growth factor A (VEGF) in a conjugation of alginate-heparin (MP-VEGF), as heparin increases the bioaffinity for the growth factor, and co-encapsulated gentamycin-loaded nanoparticles in the resulting microparticles. In a 3D microenvironment used to test the impact of MP-VEGF on angiogenesis, more endothelial sprouts with larger size were observed in MP-VEGF samples than in the free VEFG ones. In addition to the growth factor, microparticles were capable of sustainedly release gentamicin, an antibiotic incorporated into nanoparticles to inhibit bacterial growth. Microparticles were effective in carrying active substances in a formulation capable of promoting more than one activity at the same time.

Chronic wounds, such as lesions that affect diabetic people, are a challenge for researchers who try to develop an effective therapeutic alternative to rapidly heal them and reduce infectious complications. Eroğlu et al. [87] developed resveratrol microparticles incorporated in hyaluronic acid, a component of the extracellular matrix that contributes to skin integrity by stimulating cell proliferation and increasing mobility, and dipalmitoylphosphatidylcholine (DPPC), a phospholipid with structure similar to that of cell membranes. The microbiological test did not detect microbial growth, while cytotoxicity test showed no toxicity to cells, which, on the contrary, proliferated. The formulation showed effective cell growth and prolonged delivery profile suitable for the treatment of chronic wounds, which makes it a promising alternative for their treatment.

#### 3.6.2. In Vivo Healing Activity

Burns are one of the most complex and painful physical lesions to treat and manage. Hydrogels have favorable characteristics for wound healing, as they are biocompatible, flexible, mechanically resistant and semipermeable, and are available in the form of three-dimensional polymer network or spreadable viscous gel (Figure 6A,B) [88]. However, they have some drawbacks including limited drug amount that can be incorporated into the system and high water content that, despite bringing similarity to biological tissues, causes them to have fast release profiles. To overcome these limitations, Ribeiro et al. [89] prepared chitosan microparticles loaded with vascular and epidermal endothelial growth factors by ionotropic gelation to be incorporated into hydrogels (Figure 6C) and evaluated wound healing. In vivo experiments, performed to evaluate their applicability in the treatment of skin burns, showed that microparticle embedded in the hydrogels allowed a better and faster healing process than the growth factors alone.

In another study, Pereira et al. [90] prepared Aloe vera/vitamin E/chitosan microparticles to treat skin burns. Rats treated with microparticles showed increased activity of fibroblasts and keratinocytes, due to a prolonged time of formulation retention in skin lesion without significant distribution in the bloodstream, which was confirmed by gamma scintigraphy.

Lipoxin A4 (LXA4) is an eicosanoid derived from the metabolism of arachidonic acid capable of binding to various cell types, such as macrophages and neutrophils, inducing the reduction of proinflammatory functions and assisting wound healing. Being it so, Reis et al. [91] microencapsulated LXA4 in PLGA to improve its stability and tested microparticles (LXA4-MPs) in rat ulcers to evaluate their ability to accelerate the healing process. Seven days after injury in the animal skin, the groups treated with fibrin glue and PBS, soluble LXA4 and unloaded-MPs showed 39, 60 and 45% closure of wounds compared to their initial diameter, while the group treated with LXA4-MPs exhibited 80% closure and even full closure after 14 days. Collagen deposition was more intense in animals treated with LXA4-MPs than with soluble LXA4. Unlike groups treated with LXA4 and unloaded MPs, that treated with LXA4-MPs exhibited massive neovascularization and increased VEGF level on the last day, which proves the best healing action of the microparticulate system compared to free active principles.

Simvastatin is an antihypertensive drug, inhibitor of β-hydroxy β-methylglutaryl-CoA reductase, for which stimulation of VEGF, angiogenesis and endothelial function activities have been reported. Being such functions crucial for wound healing, Yasasvini et al. [92] prepared simvastatin and chitosan microparticles in polyvinyl alcohol (PVA) hydrogels to obtain a controlled release system for topical application capable of accelerating the healing of rat excision wounds. Using three different doses of simvastatin (2.5, 5 and 10 mg) in PVA microparticles, it was observed that the swelling index and the percentage of drug released within 7 days (92, 60 and 36%, respectively) were inversely dose-dependent. The histopathological study showed that granulation tissue had complete epithelialization in animals treated with low drug dose. These results suggest that a dose increase led to an increase in polymer reticulation, impairing the drug controlled release.

Wound healing in diabetic patients tends to take longer than in healthy individuals because the inflammatory phase is more prolonged, causing the release of cytotoxic enzymes, inflammatory mediators and ROS responsible for continuous damage to tissues. Therefore, Gokce et al. [7] produced a three-dimensional dermal matrix composed of collagen and laminin, to which resveratrol-loaded microparticles made up of hyaluronic acid and DPPC were added (DM-MP-RSV). A synergistic effect of the combination of dermal matrix components and the antioxidant effect of resveratrol was observed. In vivo experiments with diabetic rats showed that wounds of the group treated with DM-MP-RSV completely healed in 14 days, while those of groups not treated or treated with resveratrol were still open. The group treated with DM-MP-RSV also enjoyed more efficient re-epithelialization and larger number of collagen fibers compared to the other groups. In short, the formulation had effective and higher activity than the isolated compounds.

## 4. Current Market

Over the years, microparticulated systems have been developed with different functionality in the areas of engineering, food, cosmetics, biotechnology and pharmaceuticals. A report published by Grand View Research in 2016 entitled “Microspheres Market Size, Share & Trends Analysis Report by Type (Hollow, Solid), by Material (Glass, Polymer, Ceramic, Fly Ash, Metallic) by Application (Construction Composites, Paints & Coatings, Healthcare, Cosmetics, Oil & Gas, Automotive) and Segment Forecasts”, estimated for the microspheres market a value of USD 3.2 billion in 2015, with an expected annual compound growth rate of 10.4% (2014–2025) [93], and pointed to the health and biotechnology sector as the one in the strongest expansion.

In 2019, Lengyel et al. cited some of drugs used in clinical practice that have in their composition microparticulate systems in different pharmaceutical forms (tablets, powders, suspensions, injectable solutions, creams and gels, among others) (Figure 7) [3]. For some of these products, studies were performed that compared their effectiveness with those of conventional formulations and evaluated the pharmacokinetics, safety and effectiveness, among others; they are described below.

### 4.1. DepoDur*™*

DepoDur™ is a formulation that uses DepoFoam™ (SkyePharma, San Diego, CA, USA) for prolonged morphine sulfate release. In this technology, the drug is encapsulated in multivesicular liposomes, where each particle is formed by a set of compacted and non-concentric vesicles [94]. A study done on 70 women submitted to elective cesarean section, 35 of which received conventional morphine (4 mg) and the remaining DepoDur™ (10 mg), revealed a 40% reduction in the need for supplementary medication for pain and less limitation on mobility 48 h after cesarean section in the group treated with DepoDur™. One of the study limitations is the fact that no equianalgesic dose was determined and that doses were selected based on the analgesic ceiling from other studies, but, based on the profile of side effects and analgesia, which were similar in both groups, it is likely that doses were close to the equianalgesic one. However, DepoDur™ seemed to offer superior prolonged analgesia than the conventional morphine treatment [95].

### 4.2. Afrezza*™*

Injectable insulin treatment is the most used to treat type 1 diabetes mellitus (DM1) and, in many cases, type 2 diabetes mellitus (DM2); however, factors such as pain of application and inconvenience associated with injections are among those contributing to low adherence to this treatment. Therefore, other insulin administration routes have been studied as therapeutic alternatives since the hormone discovery. Afrezza™, or Technosphere Insulin (TI), is an inhalation powder formulation that came to the market as an alternative to injectable insulin [96,97]. In an open randomized study, Rave et al. [98] compared the use of regular insulin with that of TI, for two 7 days periods, in a group of 16 patients instructed to maintain their normal diet and exercise routines. The results showed that the level of postprandial glucose (30–120 min) in patients treated with IT was lower than in those treated with regular insulin, with AUC values from 0 to 240 min 52% lower.

Rosenstock et al. [99], in a double-blind, randomized, multicenter, placebo-controlled study, compared the effectiveness, tolerability and safety of TI with that of Technosphere Placebo powder for 12 weeks in 126 patients who had never used insulin but were doing oral therapy against DM2. At the end of the study, the level of glycated hemoglobin in the group of patients treated with TI was reduced by 56% compared to the beginning of treatment and was 43% lower than in the placebo-treated group. In another study, the same research group [100] compared blood glucose levels for 52 weeks in 654 patients, 323 of which had received TI and insulin glargine and 331 70/30 insulin aspart (IA) in controlled doses and number of applications. The glycated hemoglobin level was similar in both groups, but the fasting and postprandial blood glucose levels were lower in the TI-treated group.

Acceptance studies have also been carried out on insulin inhalation. Peyrot and Rubin [101] investigated the effects of IT and IA for 45 weeks and, as a secondary outcome, patient satisfaction data were collected through the 36-Item Short Form Survey (SF-36). Physical and mental health scores of patients in the IA-treated group decreased compared to the beginning of treatment, while there was no significant reduction in those of patients of the IT-treated group. In addition, whereas patients of the IA-treated group reported increased body pain levels, those of the IT-treated group showed greater satisfaction with treatment and improved behavior in relation to it.

### 4.3. Betaloc*™* ZOK

Egstrup et al. [102], in a randomized, double-blind, cross study, compared metoprolol CR (Betaloc™ ZOK, controlled-release microparticle formulation) administered once a day with a conventional metoprolol formulation administered twice daily. For the study, 115 patients with stable chest angina were selected and treated with 100 and 200 mg/day microparticles, according to the β-blocker dose previously used. Although the antianginal effect was similar in both groups, the one treated with 200 mg/day metoprolol CR showed greater tolerance to exercise, as chest pain began later. In groups treated with 100 mg/day, patients that received a single dose of microparticles had fewer adverse effects than those treated with two 50 mg doses of conventional formulation, indicating better tolerance. In addition, subjective symptoms were reported in a smaller quantity in the metoprolol CR-treated group.

Oosterhuis et al. [103] compared the pharmacokinetic and pharmacodynamic profile of two metoprolol formulations, i.e., metoprolol CR and metoprolol SR (traditional slow-release formulation), in a randomized, simple, placebo-controlled study. The pharmacodynamic study showed that the formulations had similar β1-blocker effect, but the effect between doses was different, with metoprolol CR showing lower peak-trough fluctuation ratio and keeping plasma concentration >75% of *C*_max_ longer. The difference between the *C*_max_ and *C*_min_ values was lower for metoprolol CR, which maintained plasma concentrations at more constant values for longer times.

### 4.4. Lupron Depot*™*

Leuprolide acetate is a synthetic agonist of Luteinizing Hormone Release Hormone (LHRH) that induces negative LHRH receptor regulation, causing reversible chemical castration. To check the effectiveness and safety of Lupron Depot™ (leuprolide acetate for depot suspension, 3.75 mg), Marberger et al. [104] conducted a multicenter phase III study, in which the drug was injected monthly in 106 men with prostate cancer diagnosis for 6 months. Blood samples for testosterone, LH hormone and follicle-stimulating hormone dosage were collected throughout the study, while the safety profile was evaluated through laboratory tests, including hematology, coagulation parameters and clinical chemistry. Blood pressure and heart rate measurements, as well as electrocardiograms, were also made. The pharmacokinetic study evidenced a prolonged release profile of the drug, which could be detected up to 28 days after the first administration. After this time, 96.8% of patients reached testosterone levels corresponding to chemical castration, and, by the end of the study, all who completed treatment (*n* = 152) maintained these levels in the following monthly evaluations. The testosterone suppression profile was similar to that observed by Perez-Marreno et al. [105], who administered another leuprolide acetate formulation subcutaneously every 28 days but in a double dosage (7.5 mg/day). Figure 8 illustrates the way Lupron Depot^TM^ disintegrates by the erosion of microparticles.

Finally, the production and development of microparticles are still a field of continuous and promising research, with products inserted in the market that prove the effectiveness of this technology and its advantages over conventional release systems. Table 2 lists the main products and summarizes the benefits each brings.

## 5. Conclusions

Based on exposed studies, it is possible to state that the insertion of active pharmaceutical ingredients in microparticles is effective in enhancing their main characteristics, such as solubility, bioavailability, controlled release and so on, which, in turn, can reduce the number of administrations and drug adverse effects. In addition, microparticles can protect such active principles from aggression from external agents and enable them to reach intracellular targets due to the improvement of their permeability. Microparticles are being developed and studied through both in vitro and in vivo tests involving the most diverse biological activities, such as the anticancer, antiparasitic, antimicrobial, antioxidant and anti-inflammatory ones. The presence on the current market of some drugs based on this delivery technology proves its promising potential. For the development of microparticles, some factors are considered of great relevance, including polymeric matrix selection and preparation methods. In this sense, biocompatible and inert polymers are preferable, as they reduce the chance of adverse effects. Moreover, some of them even offer synergistic effects with that of the active principle, such as the chitosan matrix, which, in several studies, has shown to have antibacterial activity. Microparticles can be obtained employing either drugs already available on the market, which have attributes to be improved, or new, active pharmaceutical ingredients not yet available as drugs. Drugs of this type already available commercially reinforce the wide application and the positive impact of microscale technology on the pharmaceutical industry.

## Figures and Tables

**Figure 1 ijms-24-05441-f001:**
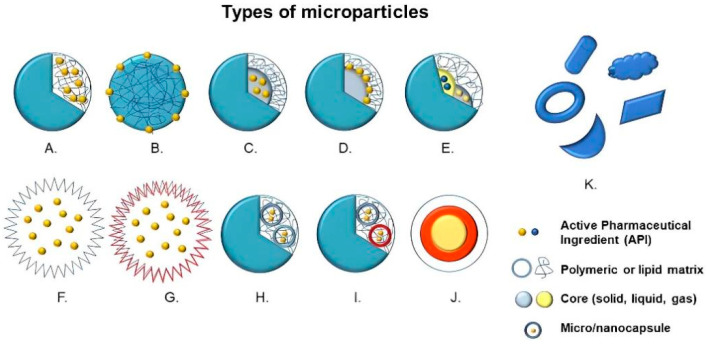
Types of microparticles: (**A**) Microsphere with entrapped active pharmaceutical ingredient (API), (**B**) Microsphere with adsorbed API, (**C**) Microcapsule with entrapped API, (**D**) Microcapsule with adsorbed API, (**E**) Multinucleated microcapsule, (**F**) Hollow microparticle, (**G**) Hollow microparticle with several layers, (**H**) Microparticle containing microcapsules, (**I**) Microparticle containing multinucleated microcapsules, (**J**) Multilayer microparticles and (**K**) Microparticles with irregular shapes.

**Figure 2 ijms-24-05441-f002:**
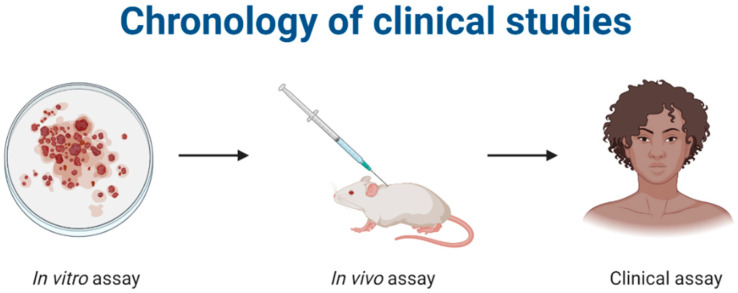
Representation of the stages of a clinical study on bioactive compounds. Created with Biorender.com, accessed on 20 March 2022.

**Figure 3 ijms-24-05441-f003:**
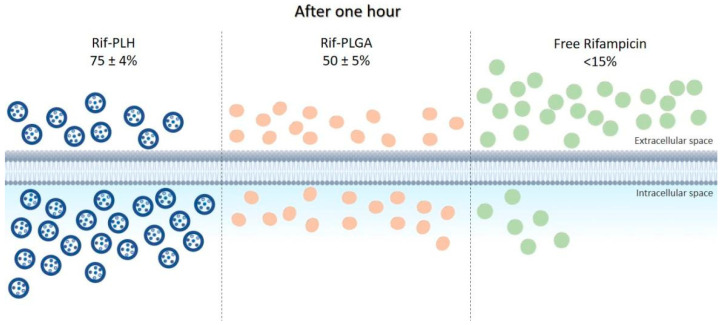
Intracellular uptake of the Rif-PLH, Rif-PLGA and free drug systems after one hour of incubation in cell culture.

**Figure 4 ijms-24-05441-f004:**
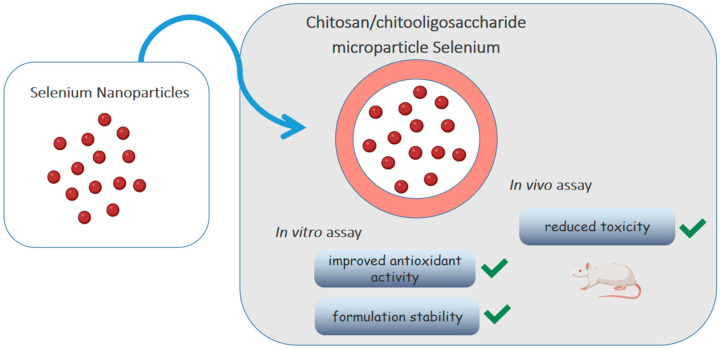
Chitosan/chitooligosaccharide microparticles carrying selenium nanoparticles.

**Figure 5 ijms-24-05441-f005:**
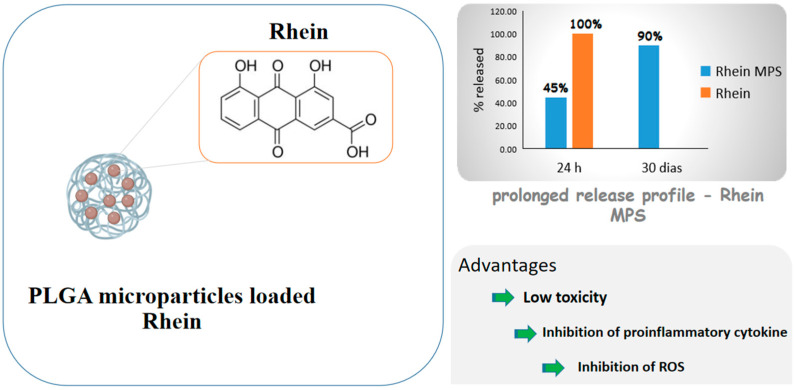
Scheme depicting rhein-carrying poly(lactic-co-glycolic acid) (PLGA) microparticles and drug advantages after encapsulation.

**Figure 6 ijms-24-05441-f006:**
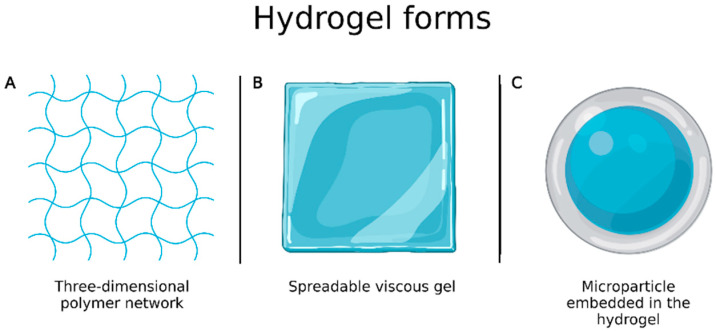
(**A**) Hydrogel in the form of three-dimensional polymer network; (**B**) hydrogel in the form of spreadable gel; (**C**) microparticle embedded in the hydrogel. Created with Biorender.com. accessed on 20 March 2022.

**Figure 7 ijms-24-05441-f007:**
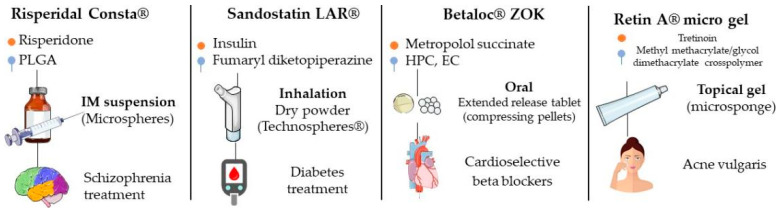
Examples of products with indication of drug, main excipient, administration form, type of microparticle and clinical indication.

**Figure 8 ijms-24-05441-f008:**
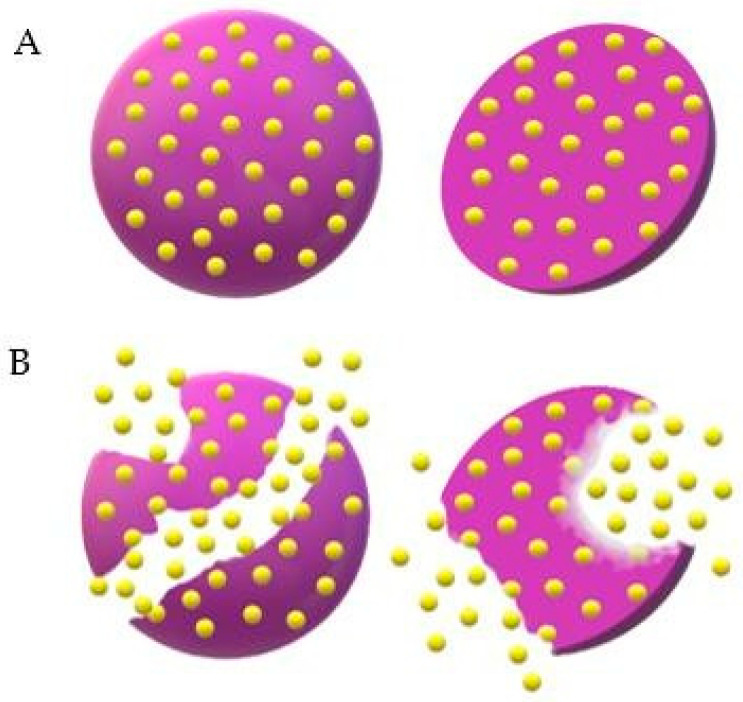
Illustration of drug release from Lupron Depot™ microspheres. (**A**) Intact microspheres loaded with the drug, represented in yellow, dispersed homogeneously within the matrix. (**B**) Beginning of erosion with drug release in the medium.

**Table 1 ijms-24-05441-t001:** Studies in which chitosan microparticles were used as a single active ingredient or to act in synergy with other compounds with antibacterial activity.

Pathogen	Active Principle	Type of Study	Particle Size (µm)	**Ref.**
*Escherichia coli* O157:H7 EDL933 (ATCC48935), Intrauterine pathogenic *E. coli*, *Salmonella enterica* CDC3041-1, and *Klebsiella pneumoniae*	Chitosan	In vitro and in vivo	0.6 ± 0.076	[54]
*E. coli*, *Pseudomonas aeruginosa* and *Staphylococcus aureus*	Ciprofloxacin and chitosan	In vitro	0.712 ± 122 0.720 ± 153	[55]
*S. aureus*, *E. coli*, *P. aeruginosa* and *Bacillus subtilis*	Selenium	In vitro	0.592 ± 0.057	[56]
*Streptococcus mutans*	Chitosan	In vitro	5.61	[57]
*Fusobacterium necrophorum* and *Bacteroides pyogenes*	Ceftiofur	In vitro, in vivo and ex vivo	-	[58]
*Pseudomonas fluorescens*, *Erwinia carotovora* and *E. coli*	Chitosan	In vitro	0.06 ± 5.48; 0.078 ± 6.77; 0.105 ± 8.58	[59]
*Vibrio cholerae*	Chitosan	In vitro	0.6 ± 0.076	[60]
Multi-drug resistant (MDR) coagulase-negative *Staphylococcus*, MDR *Pseudomonas* p41, MDR *Pseudomonas* p21, *S. aureus* ATCC 6538 and *P. aeruginosa* ATCC 9027	Copper oxide, Tetracycline and Chitosan	In vitro	1	[61]
*Salmonella enterica*	Chitosan	In vitro	-	[62]

**Table 2 ijms-24-05441-t002:** Products present in the market that use polymeric microparticles as release system, their respective active principles, polymers used in the formulation and advantages that each formulation brings.

Product	Active Principle	Matrix	Advantages	Ref.
DepoDur™	Morphine	Cholesterol, DOPC, DPPG, tricaprylin, triolein	Need for lower dose to obtain the therapeutic effect; less adverse effects.	[94,95]
Afrezza™	Insulin	Fumaryl diketopiperazine	Noninvasive administration; better blood glucose control.	[99,101]
Betaloc™ ZOK	Metoprolol succinate	Hydroxypropyl cellulose	Less adverse effects; maintenance of drug plasma concentration for longer time.	[102,103]
Lupron Depot™	Leuprolide acetate	PLGA	Need for lower dose to obtain the therapeutic effect; prolonged release.	[104,105]

DOPC: 1,2-Dioleoyl-sn-glycero-3-phosphocholine; DPPG: 1,2-Dipalmitoyl-sn-glycero-3-phospho-rac-(1-glycerol); PLGA: poly(lactic-co-glycolic acid).

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
