# Peer review of "Microparticles in the Development and Improvement of Pharmaceutical Formulations: An Analysis of In Vitro and In Vivo Studies"

_ijms, 2023, doi:10.3390/ijms24065441_

Round 1

Reviewer 1 Report

Microparticles have been an important area for industry and academia for decades. This review focuses on the recent progress and give s a detail information about the filed. In general, this paper is well-written and easy to follow. There a few suggestions.

The 3.1.3, the title of this section is “In vivo anticancer activity”, which seems respectively to the 3.1.1 and 3.1.2.

The review will need to discuss “exploring features improved in bioactive compounds by microencapsulation such as their solubility and physicochemical properties,” as suggested in lines 72-76. However, there is not enough exploration of why the compound was formulated into microparticles and how the formulation can improve its physicochemical properties.

Reviewer 2 Report

The authors presented a review on in vitro and in vivo studies considering different drug-delivery microparticles. They referred to 105 references. They also claimed that "Because of their anatomical differences, animal models may not reflect reliably what occurs when humans are exposed to these particles [8]." The concentrated information will be of help to those entering into the field and also to those following the latest development.

Reviewer 3 Report

This review describes the microparticulate systems as a tool for functional biomaterials such as drug delivery systems and biological activity assays. Since the approach described in this review are a useful for studies on microparticulate systems in the pharmaceutical field, I consider the paper acceptable for publication at International Journal of Molecular Sciences after addressing the comments and thorough revision of the manuscript.

Comments

The manuscript is well discussed with the citing literature, but it is unfriendly for the reader to understand the specific examples because there are no figures including the molecular structure and schematic diagram (especially, 3.1-3.5.). I would like you to add figures as appropriate to help the reader understand.
